# Malian field isolates provide insight into *Plasmodium malariae* intra-erythrocytic development and invasion

Francois Dao[1,2], Amadou Niangaly[1], Fanta Sogore[1], Mamadou Wague[3], Djeneba Dabitao[3], Siaka Goita[1], Aboubacrin S. Hadara[1], Ousmaila Diakite[1], Mohamed Maiga[1], Fatoumata O. Maiga[1], Chantal Cazevieille[4], Cecile Cassan[2], Arthur M. Talman[2], Abdoulaye A. Djimde[1], Alejandro Marin-Menendez[2�]*, Laurent Dembélé[1☯]*

**1** Malaria Research and Training Center (MRTC), Université des Sciences, des Techniques et des Technologies de Bamako (USTTB), Bamako, Mali, **2** MIVEGEC, Université de Montpellier, CNRS, IRD, Montpellier, France, **3** Faculty of Pharmacy and Faculty of Medicine and Odonto-Stomatology, University Clinical Research Center (UCRC), University of Sciences, Techniques, and Technologies of Bamako (USTTB), Bamako, Mali, **4** Hôpital Saint Eloi - Bâtiment INM, Montpellier, France

☯ These authors contributed equally to this work.
* laurent@icermali.org (LD); alejandro.marinmenendez@ird.fr (AMM)

## Abstract

*Plasmodium malariae* is the third most prevalent human malaria parasite species and contributes significantly to morbidity. Nevertheless, our comprehension of this parasite's biology remains limited, primarily due to its frequent co-infections with other species and the lack of a continuous *in vitro* culture system. To effectively combat and eliminate this overlooked parasite, it is imperative to acquire a better understanding of this species. In this study, we embarked on an investigation of *P. malariae*, including exploring its clinical disease characteristics, molecular aspects of red blood cell (RBC) invasion, and host-cell preferences. We conducted our research using parasites collected from infected individuals in Mali. Our findings revealed anaemia in most of *P. malariae* infected participants presented, in both symptomatic and asymptomatic cases. Regarding RBC invasion, quantified by an adapted flow cytometry based method, our study indicated that none of the seven antibodies tested, against receptors known for their role in *P. falciparum* invasion, had any impact on the ability of *P. malariae* to penetrate the host cells. However, when RBCs were pre-treated with various enzymes (neuraminidase, trypsin, and chymotrypsin), we observed a significant reduction in *P. malariae* invasion, albeit not a complete blockade. Furthermore, in a subset of *P. malariae* samples, we observed the parasite's capability to invade reticulocytes. These results suggest that *P. malariae* employs alternative pathways to enter RBCs of different maturities, which may differ from those used by *P. falciparum*.

## Author summary

Malaria is a parasitic disease caused by *Plasmodium* parasite genus transmitted by *Anopheles* mosquitoes. Five different species are involved in human infection, and two of them have been largely studied, *P. falciparum* and *P. vivax*. However, within an eradication

**Data availability statement:** All data produced in this study and used for analysis is available in the Supporting information file.

**Funding:** The project training activities in France were funded by the French Embassy in Mali to L.D. and F.D., while the field and lab research in Mali were supported by the Science For Africa (SFA) Foundation though Grand Challenge Africa (Reference GCA/DD2/Round10/030/004 to L.D. and A.D.D. A.M.M. is a recipient of a H2020-MSCA Individual Fellowship (891456). Internal funding for experiments was provided from the French Institute for Sustainable Development (IRD) to A.M.T. and A.M.M. The funders had no role in study design, data collection and analysis, decision to publish, preparation of the manuscript.

**Competing interests:** The authors have declared that no competing interests exist.

agenda in mind, the other three less explored species should not be neglected. In this study, we focused on the investigation of one of this species, *P. malariae*, for which little is known, mainly due to the lack of a continuous in vitro system and the frequency of co-infections with other species. Here, we studied the clinical manifestations of the infections, optimised a flow cytometry-based method aiming to decipher molecular interactions between the parasite and the red blood cells, the main parasite host-cell. Our results indicate that *P. malariae* might use different invasion pathways than other *Plasmodium* species, including the possibility of invading red blood cells of distinct maturities.

## Introduction

There are areas in forest and wet savannahs of West and Central Africa where prevalence of *P. malariae* may represent up to 10–20% [1–4] although, worldwide, *P. falciparum* and *P. vivax* account for the majority of cases of malaria. Over the last decade, the World Health Organisation has defined various strategies to eliminate malaria. However, *P. malariae* has clearly been neglected having been barely mentioned in the latest report [5].

Five species of malaria parasites (*P. falciparum*, *P. vivax*, *P. malariae, P. ovale* and *P. knowlesi)* are known to infect humans [6]. Most malaria-related deaths are caused by *P. falciparum*, but severe or chronic malaria cases have also been described with *P. vivax*, and to a lesser extent with *P. malariae*, *P. knowlesi* and *P. ovale* [7]. A likely reason for the more harmful effects of *P. falciparum* is, first, that they sequester in deep vascular microcirculation causing cerebral malaria. Secondly, *P. falciparum* parasites invade and grow in RBCs of all maturities and their development ends with bursting the RBCs which can lead to severe anaemia [8,9]. *P. malariae* rarely progresses to life-threatening episodes. However, glomerulonephritis or severe anaemia have been consistently reported in children with chronic *P. malariae* infections [10] leading to substantial impacts on morbidity.

All *Plasmodium* species infecting humans share a similar life cycle [11]. The *Plasmodium* life cycle inside of the human body, begins when parasites enter the blood following a mosquito bite [12]. These parasitic forms rapidly reach the liver and invade hepatocytes, where they proliferate by thousands [13]. The resulting parasite forms are released back into the blood and infect RBCs. In a RBC, one merozoite multiplies asexually to generate, depending on the species, between 8 and 64 new merozoites [14]. These new merozoites break the host-cell and this intraerythrocytic propagation cycle is repeated every 24 (*P. knowlesi*), 48 (*P. falciparum*, *P. ovale*, *P. vivax*) or 72 (*P. malariae*) hours. Some merozoites then differentiate into the next developmental stage, called gametocyte, for sexual reproduction [15]. Following a mosquito blood meal, in the lumen of the mosquito midgut, the male and the female gametes fuse in the midgut where the parasite cycle continues [16].

In recent years, there have been significant advancements in our understanding of the natural sequence of molecular events involved in invasion by both *P. falciparum* and the human host's RBCs [17]. These insights have paved the way for the development of invasion-blocking inhibitors and antibodies that offer protection against clinical disease [18–21]. Most of the *Plasmodium* receptors that are known to be involved in RBC binding and invasion can be classified into two families. The Duffy binding-like (DBL) family includes the *P. vivax*/*P. knowlesi* Duffy-binding proteins [22,23] and the *P. falciparum* erythrocyte binding-like proteins (mainly EBA-140, EBA-175 and EBA-181) [24]. Then, the reticulocyte binding protein (RBP) family, includes the *P. yoelii* 235-kDa rhoptry proteins, the *P. vivax* reticulocyte-binding proteins (PvRBPs), and the *P. falciparum* reticulocyte homology proteins (PfRH1, PfRH2a,

PfRH2b, PfRH4, and PfRH5) [24–26]. It is well-documented that *P. falciparum* employs multiple parasite ligands, and this redundancy allows the parasite to utilize alternative pathways for RBC invasion. While some of these interactions have been extensively studied, the complete repertoire of parasite-ligand interactions remains to be fully identified and comprehensively defined [17]. Among these ligands, only the interaction PfRH5-Basigin has been confirmed as essential for *P. falciparum* invasion [27,28]. Another crucial host factor for *P. falciparum* invasion is CD55, identified in a forward genetic screening study where all tested parasite isolates failed to attach to CD55-null erythrocytes [29]. However, the parasite-ligand for this receptor has not yet been described.

As for *P. vivax*, the most prevalent species outside of the African continent, there are three well-documented invasion ligand/host receptor pathways. First, the *P. vivax* DBL protein binds to the Duffy antigen receptor (CD234) and is crucial for *P. vivax* reticulocyte (younger RBCs) invasion [30]. Secondly, the *P. vivax* RBP2b and PvRBP2a interact with two reticulocyte markers, the transferrin receptor 1 (TRF1/CD71) and CD98, respectively [26,31–34]. Recently, it has been found that all *P. vivax* strains utilize both DARC and CD71, but with significant variation in receptor usage, which suggests *P. vivax*, like *P. falciparum*, might also use alternative invasion pathways [35]. Also, the mechanism of entrance into RBCs by *P. knowlesi* has been described recently, taking advantage of its unique biological characteristics [36]. However, *P. malariae* and *P. ovale* entry into RBCs has not been investigated in detail yet [37,38].

The genome sequence of *P. malariae* [39] revealed a family of heterodimeric proteins (fam-l and fam-m) found to have significant structural similarities to PfRH5. One of those, the RH5 kite-shaped fold, is known to be also present in RBP2a in *P. vivax* and may be a key conserved structure for the binding capabilities of all Rh and RBP genes [31,32,40]. In addition, the host adaptation in the *P. malariae* lineage has been described to be less restrictive than in *P. falciparum*, as the former can infect not only humans but also a range of primate hosts [41].

Finally, *P. falciparum* has been suggested to invade both normocytes and reticulocytes, which are characterized by the expression of CD71 in the surface [42–44]. On the other hand, *P. vivax*, like *P. ovale,* only invade reticulocytes not normocytes. In other non-human affecting species, *P. berghei* can invade reticulocytes with higher preference than normocytes [45–47] and the simian malaria parasite *P. cynomolgi* invades reticulocytes predominantly [48,49]. Interestingly, lethal strains of *P. yoelii*, another rodent parasite, invade both mature and immature RBCs, while nonlethal strains preferentially invade reticulocytes [50]. Several decades ago, *P. malariae* was suggested to have a preference for normocytes observed by microscopy [51]. However, the debate has recently been reopened [52] and, to our knowledge, no recent advances have explored further the molecular interplay between this parasite and the host RBCs using more quantitative technologies, such as flow cytometry [53].

*P. malariae* natural infections are characterized by low parasitaemia and recurrent co-infections with other *Plasmodium* species. Together with the absence of long-term continuous *in vitro* culture, the knowledge on this species is still scarce. In this study, we aimed to provide a better understanding of *P. malariae* biology, behaviour within the host and its interaction with the human RBCs using field isolate *P. malariae* parasites collected from infected individuals in Mali.

## Materials and methods

### a. Ethical approval

Prior the start of the activities, the study protocol was approved by the ethical committee of the Faculty of Medicine and Odonto-Stomatology, and the Faculty of Pharmacy of the

University of Sciences, Techniques and Technologies of Bamako (USTTB) with the reference Nº2020-2023/168/CE/FMPOS/FAPH renewed in 2023. The non-infected blood samples came from the Malian Blood Bank and the infected samples were obtained as described below in section b, all complying with the Ethical protocol in place for this study.

## b. Recruitment of participants

Our study took place in Faladie, within the municipality of Ntjiba located in the constituency of Kati in the region of Koulikoro in Mali. This municipality has 23 villages with an estimated population of approximately 23,000 inhabitants, according to the local registers (RAVEC 2009). The screening survey was conducted during the low transmission season (February to March in four different villages). In this region, malaria is hyperendemic, and transmission is seasonal with the rainy-high transmission season occurring between July and December. A thick smear was taken for all children after obtaining written informed consent from any of their parents or legal guardian. All blood samples collected were anonymised. Parasitae-mia was assessed by counting the number of parasites per 200 leukocytes on Giemsa-stained thick blood smears, assuming an average number of 8000 leukocytes/µL. 45 symptomatic and asymptomatic individuals infected with *Plasmodium malariae* and who met the inclusion criteria were enrolled in our study. and a 5 mL venous sample were collected in a Citrate Dextrose Acid tube. Recruitment was carried out without knowledge of their genetic history. Positive volunteers were aged between 4 and 14 years. *P. malariae* was diagnosed at the site by stained blood smears, detected by light microscopy and 30 cases were later confirmed to be *P. malariae* monoinfections by qPCR. Co-infection with other malaria species (*P. falciparum* and *P. ovale*) were identified and excluded from the experimental study. Upon sample reception, after plasma removal and two washes with RPMI, samples were resuspended at 2% haematocrit in complete RPMI medium containing 10.43 g of RPMI-1640, 5.96 g of HEPES, 2.5 g of NaHCO$_3$, 1 mL of hypoxanthine, 5 g of Albumax, 2.5 mL of gentamicin 50 mg/mL in 1 L of H$_2$O lacking extra glucose supplement [54] and used for subsequent experiments as described below.

## c. Quantitative PCR (qPCR)

As described elsewhere [54], molecular diagnosis was performed by amplification of the *P. malariae* 18S gene including an internal control, the human actin gene, by quantitative PCR on DNA samples extracted from filter paper with a Qiagen kit. Primer and probe sequences of *P. malariae* 18S are as follows: Fw (CCGACTAGGTGTTGGATGATAGAGTAAA), Rev (AACCCAAAGACTTT-GATTTCTCATAA), probe sequence (FAM-CTATCTAAAAGAAACACTCAT). While primer and probe sequences for the human actin gene are Fw (ACCGAGCGCGGCTACAG), Rev (CTTAATGTCACGCACGATTTCC) and probe (HEX-TTCACCACCACGGCCGAGC). Each amplification solution for a single sample (10 µL final volume) consisted of 2 µL of DNA sample and 8 µL of reaction mix comprising (0.5 µL of 5 µM *P. malariae* primers and probe, 0.4 µL of 5 µM human actin gene internal control primers and probe and 5 µl of 2x PrimeTime PCR Master mix (IDT)). The following protocol was run in a BioRad CFX96 thermocycler: 95 °C for 3 minutes followed by 40 cycles of 95 °C for 15 seconds and 60 °C for 60 seconds. Samples with Ct higher than 35, which could suggest non-specific amplification, were excluded from subsequent analysis.

## d. Intraerythrocytic parasite development

Giemsa-stained smears were prepared every 12 h up to 96 h and the different parasite stages were quantified by microscopic observations. Samples were incubated in a candle jar at 37 °C

in which complete media [54] was changed once daily. To evaluate the proportion of parasites with mitochondrial activity with the passage of time, 10 µL (2% haematocrit) from eight of those parasite isolates were taken in the same timely fashion and stained simultaneously with the DNA stain SybrGreen I (SG, 1:10000 dilution, ThermoFisher, S9403) and 300 nM Mitotracker (MT, ThermoFisher, M22426) in RPMI for 30 min at 4 °C. An unstained control and stained controls with SG and MT were prepared with non-infected RBCs to establish the negative populations for both fluorochromes. After incubation, samples were washed twice, resuspended in 200 µL PBS, and analysed in a BD LSR-II flow cytometer (see all details in section l). Samples were always kept on ice. A total of 1000000 events were collected for each sample.

### e. Orthologue search

The main *Plasmodium* ligands described to be involved in RBC invasion in the two main species, *P. falciparum* and *P. vivax*, were gathered (see Results section). Then, orthologues were searched in *P. malariae* using PlasmoDB (www.plasmodb.org), a database that provides detailed information in any of the *Plasmodium* species which sequence has been annotated and belongs to the genomic resources of EuPathDB, bioinformatics research centre. Regardless of the presence/absence of orthologues in *P. malariae*, a search was performed using the Basic Local Alignment Research Tool (BLAST) for all the ligands to explore the presence of *P. malariae* matches within the top 500 list. Sensitivity (S) or resistance (R) to the different enzyme treatments or the effect of different antibodies against RBC receptors are described in the references included in the list.

### f. Enrichment of *P. malariae* late stages

Upon reception of positive samples detected by microscopy, in all samples, *P. malariae* late stages (LS) were enriched using LS magnetic columns (Miltenyi Biotec, 130-042-401) according to manufacturer's instructions. After enrichment, the 1–2 µL late stages pellet was used for the different assays described in sections g, h and i. As *P. falciparum* LS are not present in circulation, we assumed this enrichment allowed us to select for *P. malariae* LS only to use in subsequent experiments.

### g. Ultrastructural evaluation by transmission electron microscopy

LS were immersed in a solution of 2.5% glutaraldehyde in PHEM buffer (1X, pH 7.4) overnight at 4 °C. They were, then, rinsed in PHEM buffer and post-fixed in a 0.5% osmic acid + 0.8% potassium Hexacyanoferrate trihydrate solution for 2 h at dark and at room temperature. After two rinses in PHEM buffer, the cells were dehydrated in a graded series of ethanol solutions (30–100%). The cells were embedded in EmBed 812 using an Automated Microwave Tissue Processor for Electronic Microscopy (Leica EM AMW). Thin sections (70 nm; Leica-Reichert Ultracut E) were collected at different levels of each block. These sections were counterstained with uranyl acetate 1.5% in 70% Ethanol and lead citrate and observed using a Tecnai F20 transmission electron microscope at 120 KV.

### h. Host RBC invasion

Since DNA dyes combined with flow cytometry provide a rapid and high-throughput alternative to manual counting of stained smears, an existing methodology for *P. falciparum* [55] was adapted to study *P. malariae* RBC invasion *ex vivo*. Briefly, the cytoplasmic stain Cell Trace Far Red (CTFR, Life Technologies, C34564) was used at 3 µM in RPMI to pre-stain 2% haematocrit RBCs (sRBCs) from a healthy volunteer, negative to thick smear examination.

Samples were incubated under rotation at 37 °C for 2 h, washed twice with 1 mL RPMI (spin at 450 g, 2 min) and re-suspended in the original volume. They were incubated at 37 °C for 30 min more. After a last wash, they were re-suspended in the original volume. CTFR allowed to distinguish freshly added sRBCs) from already present RBCs/pRBCs in the blood samples collected. Enriched-late stages were incubated with sRBCs for 24 h or 48 h. Then, samples were incubated with 50 µL of 0.5 mg/mL Ribonuclease A (Sigma-Aldrich Co, R6513) for 1 hour, washed once with 200 µL PBS and incubated with 1x SG for 30 minutes, washed twice with 200 µL and resuspended with 200 µL PBS to run by flow cytometry (see all details in section l). A total of 500000 cells were acquired for each sample in duplicates.

### i. Effect of antibodies in invasion

Similarly, LS pellets were incubated with 2% haematocrit CTFR-stained RBCs prepared from a healthy volunteer. An unique reference concentration of 50 µg/mL was used of the following antibodies against RBC receptors known to be involved in *P. falciparum* and *P. vivax* invasion: anti-CD35 (BioLegend, 332402), anti-CD55 (Invitrogen, PA5-80438), anti-CD71 (Miltenyi Biotec, 130-124-327 (Clone REA902)), anti-CD108 (BioLegend, 376602), anti-CD147 (R&D Systems, MAB3195), anti-CD233 (IBGRL, 9439PA BRIC 6), and anti-CD235a (BioLegend, 349102). Samples were incubated for 48 h in a candle jar at 37 °C with a daily change of media. For Ribonuclease A treatment and staining, samples were treated as described in section f. For flow cytometry settings see all details in section l. As before, 500000 cells were acquired for each sample in duplicates.

### j. Effect of enzymatic treatment of RBCs in invasion

All volumes here were calculated to prepare 1.1 mL of stained RBC, which sufficed for two *P. malariae* positive blood samples and controls. RBCs at 2% haematocrit were pre-stained with 3 µM CTFR as described above. The volumes were split in 4 × 250 µL tubes (labelled A, B, C, D) and the rest (100 µL) was used for controls. After a last wash with 250 µL with RPMI, pellets were resuspended in the volumes described below using RPMI: A) 250 µL of RPMI; B) 250 µL of 20 mU/mL Neuraminidase (Sigma-Aldrich Co, N7885) in RPMI; C) 250 µL of 1 mg/mL Trypsin (Sigma-Aldrich Co, T1426); D) 250 µL of 1 mg/mL Chymotrypsin (Worthington Biochemical, LS001432). Samples were incubated under rotation at 37 °C for 1 h, washed twice with 250 µL RPMI and re-suspended in 250 µL of complete media. Stained and enzymatically treated RBCs were used within 2 h. To prepare the infected RBCs for the assay, LS pellets were resuspended in 500 µL of complete media and split in 4 × 125 µL in 1.5 mL tubes labelled pA, pB, pC, pD (4 per sample). After spinning down, the supernatant was removed and mixed with the corresponding 125 µL stained and enzymatically treated RBCs prepared above. Each 125 µL was split in 2 × 50 µL and incubated for 48 h in a candle jar at 37 °C with a daily change of media. For Ribonuclease A treatment and staining, samples were treated as described in section g. For flow cytometry settings see all details in section l. A total of 500000 cells were acquired for each sample in duplicates.

### k. Identification of *P. malariae* infected reticulocytes

50 µL of 2% haematocrit pRBCs were incubated for 30 min at 4 °C with either anti-CD71-PE (1:100 dilution, Miltenyi Biotec 130-115-029 (Clone REA902)) and 300 nM MT; or with anti-CD71-APC and SG. An unstained control and stained controls with MT, SG and anti-CD71-PE/APC antibodies were prepared along with non-parasitized RBCs to establish the negative populations for all fluorochromes. After incubation, samples were washed twice, resuspended in 200 µL PBS, and analysed in a flow cytometer (see all details in section l).

Samples were always kept on ice. We collected 1000000 events for each sample in duplicates. For both approaches, background double positive events in the controls were subtracted from the ones detected in the samples to obtain true positive values.

### l. Flow cytometry settings and statistical analysis

In all cases, samples were run in a BD LSR-II flow cytometer (BD Biosciences). SG was excited by a blue laser and detected by a 530/30 filter, while the anti-CD71PE was excited by a Yellow-Green laser and detected by a 582/15BP and both MT, CTFR and anti-CD71APC were excited by a red laser and detected by a 650/10 filter. The data collected were analysed with FlowJo v10.7 (Tree Star, Ashland, Oregon) to obtain the percentage of new invasions and the percentage of parasitaemia within each cell type (normocytes and reticulocytes). All experiments were carried out in duplicates in the number of samples indicated in the text. STATA software (version 15; Stata Corp., TX USA) was used to analyse clinical and demographic data, GraphPad Prism 9.0.0 (121) was used to make figures and statistical analysis. Parametric paired t-test was calculated to compare the effect of antibodies and enzymatically treated RBCs on *P. malariae* invasion compared to their controls. An alpha level of 0.05 was established for all tests to determine statistical significance.

## Results

### *P. malariae* infections is associated with anaemia in asymptomatic and symptomatic participants

Using clinical evaluation and blood samples from 45 *P. malariae* infected individuals, determined by microscopy, we assessed whether volunteers had symptoms in three different age groups and measured, in 28 of those individuals, the haemoglobin levels (Table 1). All participants were under 14 years old (mean = 9.6 ± 2.1) with an almost equal ratio male/female (0.96, 22/23) within the participants. We found that over 40% of the participants showed clinical symptoms (19/45) and 64.3% had anaemia (Hb level <11 g/dL). The most common symptom was headache, followed by a combination of headache and fever, while there were no severe cases in this cohort. The median parasitaemia observed was 1199 parasites/µL of blood (minimum = 1120, maximum = 5120).

After conducting qPCR analysis on DNA extracted from all samples, the results confirmed that 30 of them were *P. malariae* monoinfections whereas the remaining samples were co-infections with both *P. malariae* and *P. falciparum*. For subsequent experiments, we used subsets of these 30 samples.

**Table 1. Clinical and demographic data of the participants enrolled in this study.**

| Participants | | No symptoms | Cough | Fever | Headache | Headache + Fever | Total | |
|---|---|---|---|---|---|---|---|---|
| | | | | | | | n | % |
| **Sex** | Female | 12 | 1 | 0 | 8 | 2 | 23 | 51.1 |
| | Male | 14 | 0 | 1 | 6 | 1 | 22 | 48.9 |
| **Age** | >5 | 1 | 0 | 0 | 0 | 0 | 1 | 2.2 |
| | 5–9 | 8 | 1 | 0 | 7 | 1 | 17 | 37.8 |
| | 10–14 | 17 | 0 | 1 | 8 | 1 | 27 | 60.0 |
| **Hb level** | <11 g/dL | 14 | 0 | 1 | 1 | 2 | 18 | 64.3 |
| | >11 g/dL | 6 | 1 | 0 | 3 | 0 | 10 | 35.7 |

Hb: haemoglobin.

## Multiple stages of *P. malariae* circulating in the bloodstream

We thoroughly examined thick and thin smears of 23 samples at time of collection to measure the presence of different intraerythrocytic stages (i.e. rings, trophozoites and schizonts) (Figs 1A and S1), the average number of merozoites per schizont (Fig 1B) and the presence of gametocytes in each individual sample (Fig 1C). All three asexual stages were found circulating in blood in the tested samples as follows: schizonts were only found in eleven samples (11/23, 47.8%), while gametocytes, the sexual forms required for parasite transmission, were rarely detected (4/23, 17.4%). The number of merozoites counted per schizont was seven, as in the expected range of 6–12 for this species. Interestingly, examining one of the samples by transmission electron microscopy (TEM), we observed formation of putative knobs in the surface of all infected cells with late stages of *P. malariae*, as displayed in an example in Figs 1D and S1.

## *P. malariae* invasion and development dynamics *in ex vivo* culture

To investigate the dynamics of parasite development, we used the eluted portion after magnet purification of twenty-three patient samples, aiming to enrich early stages, incubated them under standard culture conditions and monitored them every 12 h using light microscopy (Fig 2A) and, for eight of them, flow cytometry (Fig 2B). At baseline (0 h), we distinguished heterogeneous parasite stages, predominantly trophozoites and rings. After 12 h, most of the ring stages had developed into trophozoites and the presence of schizonts was still low; while at 24 h, visualising still mostly trophozoites, the percentage of schizonts increased over 10%.

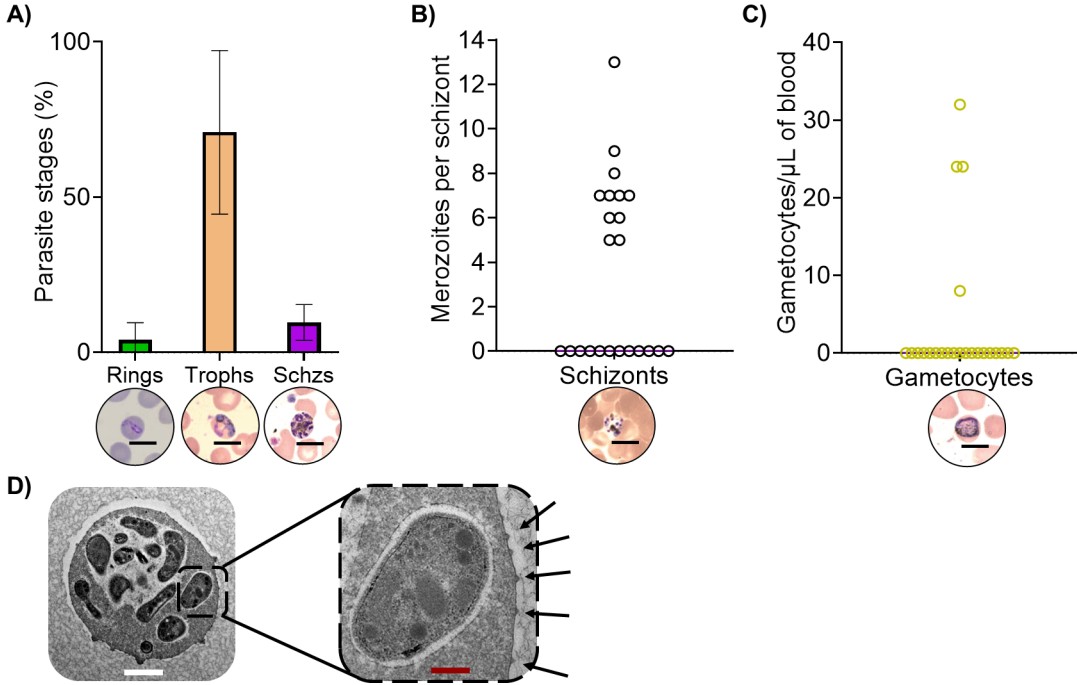

**Fig 1. Characteristics of circulating parasites in *P. malariae* natural infections.** A) Percentage of intraerythrocytic parasite stages in circulation and representative pictures of each (Rings, Trophozoites, Schizonts) (n = 23). Error bars show SD; B) Merozoites per schizont (n = 23, each point represents the average of merozoites seen in all schizonts in a given sample); C) Number of gametocytes per 1000 WBCs per sample (n = 23); D) Transmission electron microscopy image of a schizont. Black arrows indicating knobs-like structures. Black, white and red scale bar = 4, 1 and 0.2 μm, respectively.

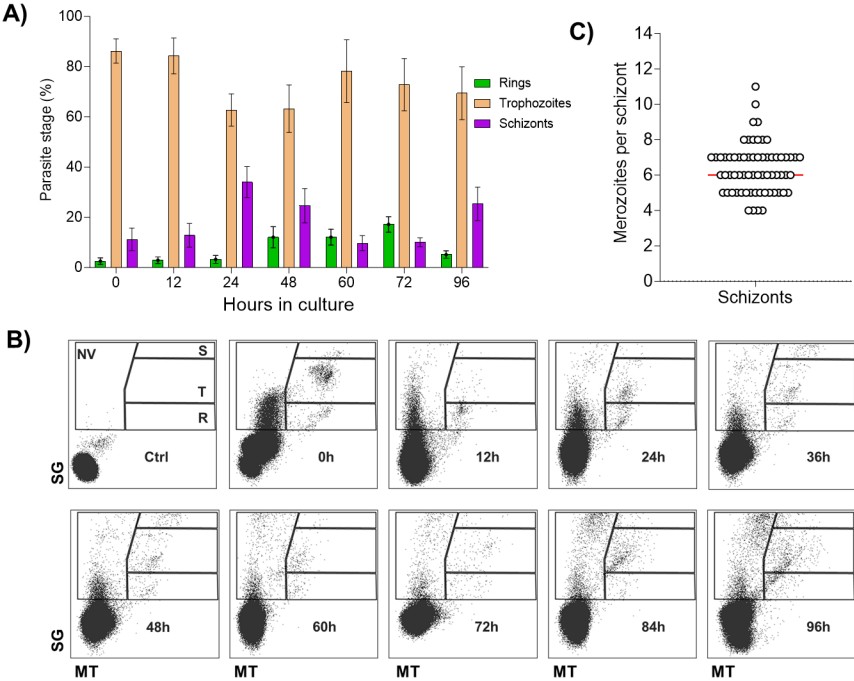

**Fig 2. *P. malariae* replicates and matures *in vitro*.** A) Percentage of intraerythrocytic parasite stages *in vitro* (from 23 samples, n = 121 exploitable smears in total for all time points. n = 23 at 0 h, n = 19 at 12 h, n = 21 at 24 h, n = 17 at 48 h, n = 14 at 60 h, n = 12 at 72 h and n = 15 at 96 h). Error bars show SD; B) Representative flow cytometry layout of parasite development within RBCs stained with SybrGreen I (SG) and Mitotracker (MT). 8 samples were timely taken every 12 hours (Ctrl: stained non-infected RBCs, NV: Non-viable; R: Rings; T: Trophozoites; S: Schizonts). The sample displayed is a *P. malariae* monoinfection confirmed by qPCR; C) Number of merozoites per schizont (n = 55 schizonts counted from 11 different samples in which schizonts were found).

At 48 h and 60 h, we observed successful reinvasions, as the reappearance of rings was evident. Finally, at 72 h and 96 h, these rings matured to trophozoite stages that dominated the culture again suggesting completion of one *P. malariae* asexual stage cycle *ex vivo* (Fig 2A).

Using SybrGreen I (SG), a DNA dye that identified parasitized RBCs (pRBCs), and Mitotracker (MT), to measure parasite viability, we observed, agreeing with the results obtained by microscopy, successful *ex vivo* reinvasion and development of *P. malariae* asexual stages by flow cytometry (Fig 2B). However, we detected an obvious accumulation of non-viable pRBCs (SG+MT-) particularly in the last two time points taken, at 84 h and 96 h, (Fig 2B) suggesting poor viability of parasites in later stages of the experiment. We observed no morphological variations by microscopy, and the number of merozoites per schizont remained unaltered compared to the number in circulating parasites (Fig 2C).

## Molecular players in *P. malariae* RBC invasion

A thorough database search has yielded a number of orthologues in *P. malariae* that have been shown to play a role in RBC invasion in other species (Table 2). None of all major *P. falciparum* known ligands of the EBL family, excluding genes annotated as pseudogenes (such as EBL1 and EBA165), have orthologues in *P. malariae* (Table 2). In contrast, all the members of the EBL family have an orthologue in *P. ovale, P. vivax* and *P. knowlesi*. In turn, three orthologues to ligands of the RBP homolog of *P. falciparum* family (PfRh1, PfRh2a and PfRh2b) have been annotated in the genome of *P. malariae*, although, their respective RBC receptors

**Table 2. Absence of *P. malariae* orthologues known *Plasmodium* ligands and RBCs receptors.**

| *Plasmodium* ligands | *P. malariae* orthologues | RBC receptor | Chymotrypsin | Trypsin | Neuraminidase | References |
|---|---|---|---|---|---|---|
| MSP1 | Yes | Band3 (CD233) | S | Nd | R | [74] |
| PfRH5 | No | Basigin (CD147) | R | R | R | [71] |
| Nd | Nd | CD55 | Nd | Nd | Nd | [29] |
| PfRH4 | No | CR1 (CD35) | S | S | R | [68,75] |
| DBP | No | DARC (CD234) | R | R | R | [76] |
| EBA175 | No | GYPA | R | S | S | [77,78] |
| EBA140 | No | GYPC | R | S | S | [59] |
| AMA1 | Yes | Kx | S | S | R | [79] |
| CyRPA | Yes | Basigin (CD147) | Nd | Nd | R | [80] |
| MSP6 | No | NA | R | R | R | [81] |
| MSP7 | Yes | NA | Nd | Nd | S | [82] |
| PfRH2a | Yes | Nd | Nd | Nd | Nd | [83] |
| RON2 | Yes | RON Complex | Nd | Nd | Nd | [84] |
| MTRAP/RIPR | No/Yes | Sema7A (CD108) | R | R | R | [85] |
| PvRBP2a | Yes | CD98 | R | S | R | [34,40] |
| PvRBP2b | Yes | Transferrin receptor 1 (CD71) | S | S | R | [31] |
| EBA181 | No | W_Band4.1 | S | R | S | [86] |
| PfRH1 | Yes | Y | Nd | R | S | [87] |
| PfRH2b | Yes | Z | Nd | R | R | [88] |

Resistant (R) or Sensitive (S) to enzymatic RBC pre-treatment; Nd: not determined; NA: not applicable as these ligands do not bind directly to the membrane.

have not yet been identified (Table 2). *P. malariae* has neither PfRH4 nor PfRH5 orthologues. Nevertheless, *P. malariae* has orthologues of the *P. falciparum* Rh5-interacting protein (PfRipr) and the Cysteine-Rich Protective Antigen (PfCyRPA) which share the PfRH5 RBC receptor, Basigin. Interestingly, the Duffy-binding protein (CD234) ligands, DBP and DBPαII, of *P. vivax* and its receptor on reticulocytes have orthologues in all other human affecting species but not in *P. malariae* (Table 2). According to these findings, *P. malariae* merozoites might rely on some orthologues of known *P. falciparum* ligands to enter the RBC, such as PfRh2b or PfRIPR, although it might as well use other ligands that are yet to be defined.

## Optimisation of an assay to quantify *P. malariae* RBC invasion and evaluate potential inhibitors

To define optimal conditions to quantify new *P. malariae* RBC invasions, we first enriched late parasite stages (LS) with magnet columns, then added pre-stained RBCs (sRBCs) with a cytoplasmic dye (Cell Trace Far Red, CTFR) and, after incubation, used SG to identify newly invaded pRBCs (Fig 3A). We quantified pRBCs both at 24 and 48 h from nine samples in duplicate (Fig 3B), determining that the latter yielded a significantly higher invasion rate in all samples tested (p = 0.02) (Fig 3C). Therefore, these settings allowed us to evaluate the impact of different potential blocking agents in the invasion process and were used for further analysis.

To assess the role of the repertoire of RBC receptors, we first used a set of seven antibodies (CD35, CD55, CD71, CD108, CD147, CD233 and CD235a), in at least four samples each, covering most of the known ligand-receptor interactions in *P. falciparum* (Table 2 and Fig 4A). At the single concentration tested (50 μg/mL), we did not observe reduction in invasion with any of these antibodies, suggesting *P. malariae* RBC entry does not rely on any of these

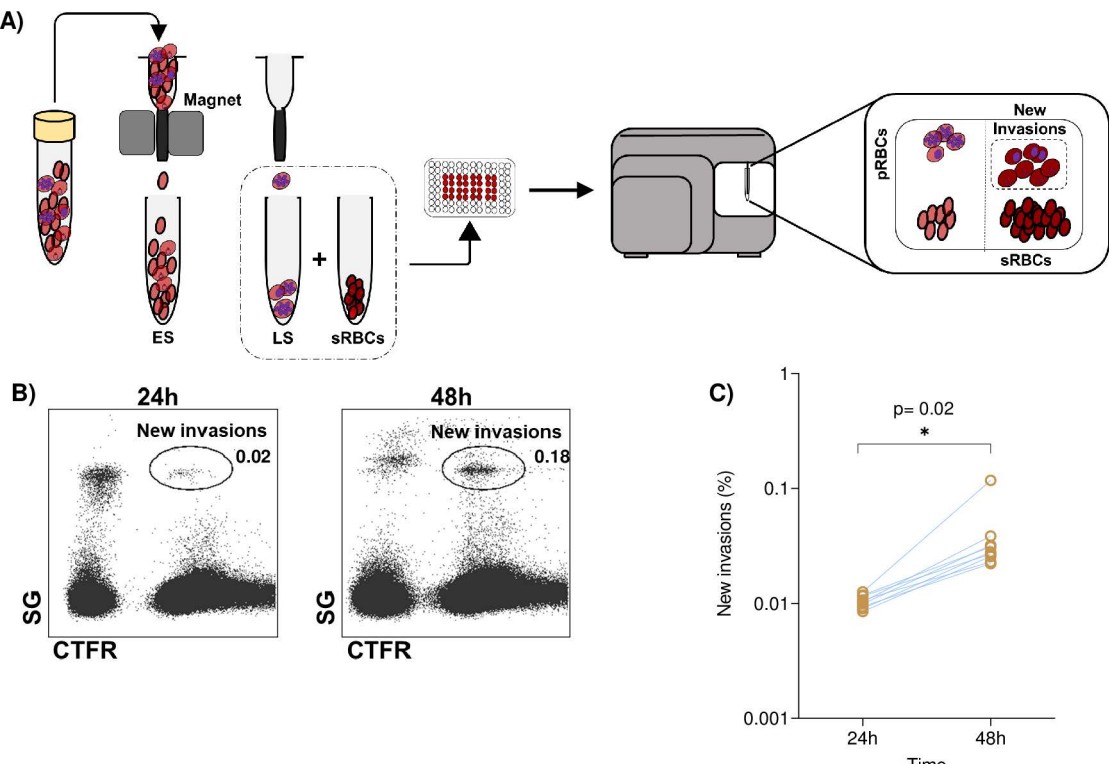

**Fig 3. *P. malariae* field isolates invade RBCs *in vitro*.** A) Experimental work plan: magnetic separation of pRBCs with early stages (ES) or late stages (LS), incubation with stained RBCs (sRBCs) and diagram of flow cytometry plots to measure new RBC invasions; B) Representative flow cytometry layout of parasite development within RBCs stained with SybrGreen I (SG) and Cell Trace Far Red (CTFR); C) Quantification of new invasions after 24 h and 48 h of *in vitro* culture (n = 9). Lines join the same sample measured at the different time points. Each dot represents the mean of the results in duplicates.

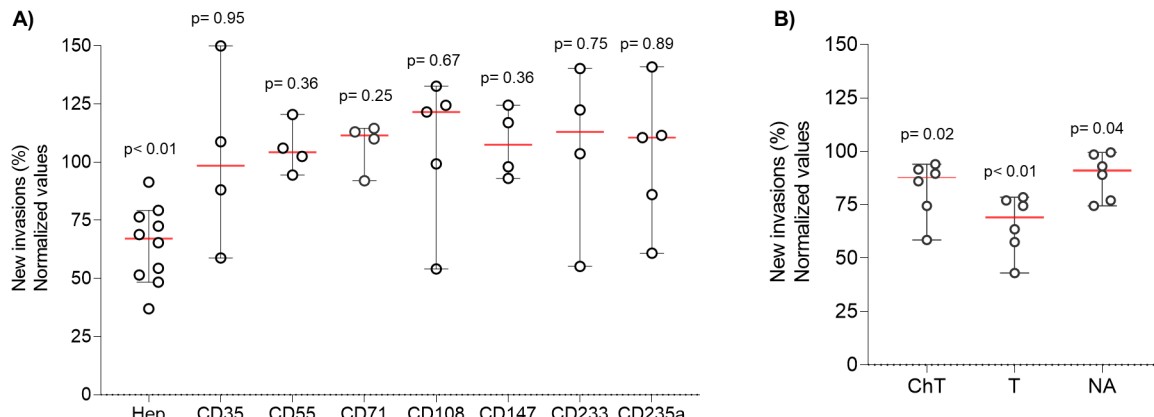

**Fig 4. Blocking agents show a partial effect on *P. malariae* RBC invasion.** A). Effect of antibodies in *P. malariae* invasion. Heparin (Hep, n = 10) was used as an inhibitory positive control. All data was normalized with a control (Ctrl) in which no antibody was used. CD35 (CR1, n = 4), CD55 (n = 4), CD71 (TR1, n = 4), CD108 (Sema7A, n = 5), CD147 (Basigin, n = 4), CD233 (Band3, n = 4), CD235a (GYPA, n = 5). B). Effect of enzymatically treated RBCs with Chymotrypsin (ChT), Trypsin (T); and Neuraminidase (NA) in *P. malariae* invasion (n = 6). Each dot represents the mean of the results in duplicates.

receptors and the merozoites of this parasite species might use alternative invasion pathways. Pre-incubation of sRBCs with chymotrypsin, trypsin, or neuraminidase, significantly blocked RBCs invasion by *P. malariae* (Fig 4B). The efficiency of the clinical isolates tested (n = 6) to invade enzymatically pre-treated sRBCs, relative to untreated cells, was reduced with all chymotrypsin (17.7%, p = 0.02), trypsin (34.3%, p < 0.01) and neuraminidase (22.5%, p = 0.04).

### Host-preference red blood cell invasion assessment

To study whether *P. malariae* displays host-cell preference, eleven samples directly from patients were analysed by flow cytometry. Samples were labelled with an anti-CD71 antibody, that allowed us to discriminate between normocytes (CD71-) and reticulocytes (CD71+), combined with either MT or SG to separate pRBCs from non-infected RBCs (Fig 5A). Representative dot-plots of both combinations CD71APC+SG and CD71PE+MT, where pRBCs and reticulocytes can be distinguished, are displayed in Fig 5B. One out of the eight *P. malariae* monoinfections tested showed invasion of reticulocytes, thus we showed for the first time, to our knowledge, the ability of this species to invade immature RBCs (Fig 5C).

### Discussion

In this study, we recruited *P. malariae* carriers from the commune of Ntjiba, a rural area of high malaria transmission in Mali, where malaria is transmitted throughout the year following seasonal variations. The prevalence of *P. malariae* is known to fall between 2 to 5% in our study area in agreement with the latest national surveys [56] and latest epidemiological data in the region [57]. Despite having been identified as *P. malariae* monoinfections by microscopy, some of our samples resulted in co-infections with *P. falciparum* after qPCR analysis. We identified all intraerythrocytic stages but only a low number of the samples examined displayed schizonts. This suggests that there might be certain cases of sequestration of late stages

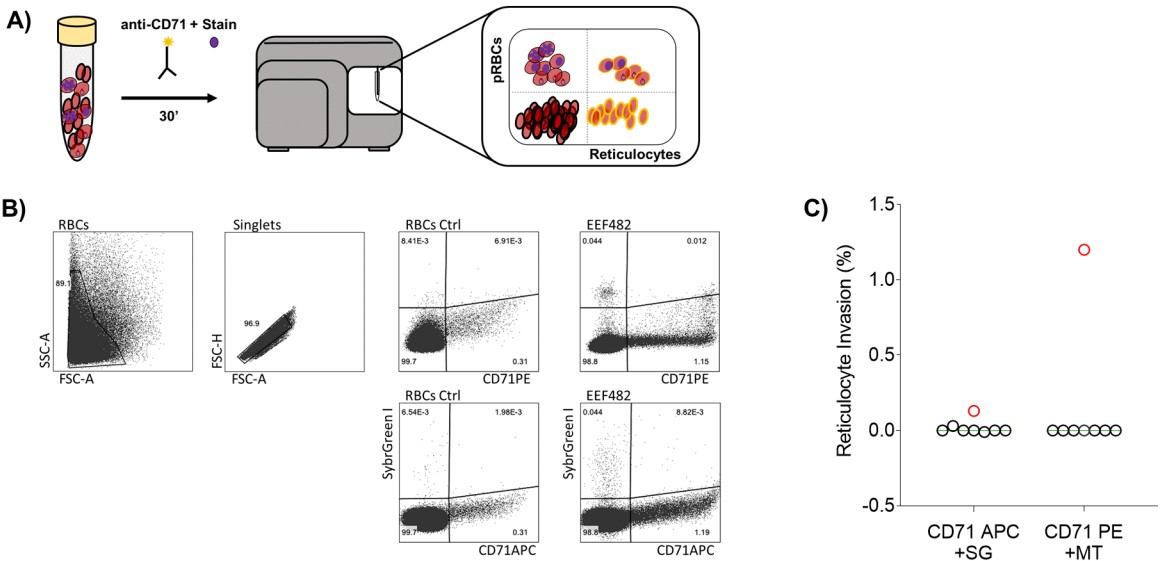

**Fig 5.  *P. malariae* is capable to invade reticulocytes.** A) Experimental workflow diagram; B) Representative flow cytometry plots with the two combinations tested (anti-CD71APC+SG; anti-CD71PE+MT). Ctrl: non-infected RBCs. EEF482: *P. malariae* monoinfection confirmed by qPCR; C) Percentage of parasitized reticulocytes (CD71+ cells). In red, same sample (EEF482) displaying successful reticulocyte invasions with both approaches (n = 8).

of infected RBCs in microcirculation as described for *P. falciparum* [58]. As we have corroborated here, the formation of knobs-like structure along the surface of *P. malariae* pRBCs was already described elsewhere, but their implications in interactions within the host were, and still are, overlooked [59]. There are a number of orthologues for several proteins associated to protein export to the RBC membrane (such as PfSBP1 or PfA66) [60], and a putative knob-associated histidine rich protein (PmUG01_02025800), a major component of knobs in the erythrocyte membrane in *Plasmodium* pRBCs, although their function remains unexplored. No *var* genes orthologues have been identified, unlike in *P. falciparum* where PfEMP1s in the RBC membrane are encoded by var genes [61,62]. Thus, the molecular composition of these knobs, and whether they play a role in sequestration of *P. malariae* infected RBCs is still unknown. In addition, we were puzzled by the low number of samples where gametocytes were present, opening the door for further research on how *P. malariae* is transmitted.

In this study, we have successfully maximised the efficiency of the first invasion cycle that can occur in laboratory conditions. During short-term culture, we have not observed any apparent abnormal parasite forms, and we have developed a reliable procedure for quantifying new invasions in this species. This system has enabled us to delve deeper into the molecular mechanisms employed by this parasite to penetrate RBCs. This invasion process hinges on the interactions between parasite ligands and RBC membrane receptors, some of which have already been identified and characterized in other *Plasmodium* species [63] but remained unexplored for *P. malariae*.

To this end, and to our knowledge, this study represents the first comprehensive investigation into not only the invasion process employed by *P. malariae* but also the ability to do so in host cells of different maturities. By optimizing an efficient flow cytometry-based method capable of handling low parasite loads and small volumes, we have undertaken the task of dissecting the intricate mechanisms used by this parasite to penetrate RBCs. Our findings indicate a reduction in invasion with each of the individual enzymes commonly tested in *P. falciparum* studies, albeit none of them resulted in a complete blockade of invasion. The availability of at least two receptors, with different enzymatic profile, in each stage of the process could make the parasites counterbalance the effect of individual enzymes. While *P. malariae* may prefer invading through a sialic acid-independent pathway, as pre-treatment of RBCs with neuraminidase had a lower effect than the other peptidases, it is likely to employ alternative mechanisms to gain entry into RBCs, as well. A model showing potential invasion pathways is shown in Fig 6. In alignment with this hypothesis, a report from the last decade described an unconventional invasion pathway utilized by Colombian and Peruvian field isolates that was resistant to simultaneous pre-treatment of RBCs with these enzymes [64]. In the same line of thought, *P. vivax* and *P. knowlesi* had been suggested to employ a Duffy-independent pathway for invasion [65] that relies on reticulocyte-binding receptors either alone or in conjunction with another unidentified parasite molecule. The current challenge lies in determining the identity of any essential receptors that have remained elusive thus far.

In contrast to enzymes, which are more prone to cleaving multiple receptors simultaneously, the utilization of specific antibodies targeting parasite receptors has yielded more dependable data regarding the function of merozoite invasion ligands [66,67]. However, our observations revealed no decrease in invasion when using seven antibodies targeting various known *P. falciparum* receptors. Despite these findings, a role for any of these receptors should not be completely discarded as, first, only a single concentration for each antibody was used and, second, these were all monoclonal antibodies, thus some of the unknown but key epitopes might be still available for the parasite. Thus, future attempts to dissect the importance of these or other putative receptors could include utilisation of polyclonal antibodies or even recombinant soluble receptors [68]. Together with the challenges posed by the inability to

genetically modify *P. malariae*, our ability to conduct a more detailed dissection of the specific receptor-ligand interactions in this parasite are hindered. In light of these results, it becomes crucial to initiate an in-depth investigation of the two large highly diverse families of genes, fam-l and fam-m [69,70]. Proteins encoded by these families show characteristics of proteins likely exported from the parasite to the pRBC surface such as a PEXEL export signal, a signal peptide [39,69] or transmembrane domains. Many of these genes encode proteins with structural homology to Rh5 of *P. falciparum*, which is the sole known protein essential for RBC invasion by *P. falciparum* [71]. Producing and assessing antibodies against some members of these families could potentially clarify their role in *P. malariae* RBC invasion.

Based on our database search findings and the attempts to block RBC invasion, we hypothesise here possible interactions that *P. malariae* establish to penetrate into RBCs (Fig 6). The initial contact could be established via either MSP1-Band3 or between an unknown ligand(s), as no EBA orthologues have been found, with any member of the glycophilin family. Then, the merozoite could reorientate apically thanks to the interplay with several receptors, such as TFR1 and CD98, and move to form a tight junction with either the RON complex and/ or coupling AMA1 with the Kx receptor. After a last interaction with both Sema7A and one of its ligands, RIPR, or of a yet-to-be-described ligand for CD55, these events would lead to invasion. Proteins for which no antibody was tested in this study but could be involved in this complex interplay are boxed.

Our study has shed light on additional characteristics of *P. malariae* within the host. Given that *Plasmodium* species can encounter RBCs at different stages of maturity in circulation, including CD71+ immature reticulocytes and more mature normocytes (CD71-), and considering the known diversity in host-cell preferences among different species, we explored

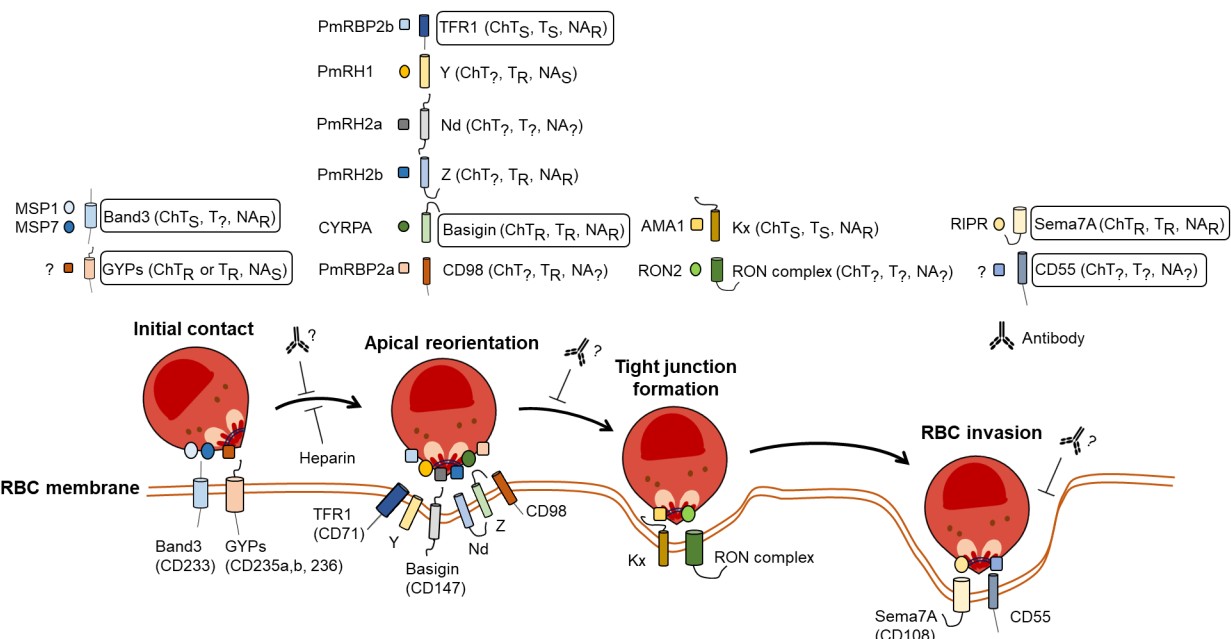

**Fig 6. Model of hypothetical *P. malariae* pathways to entry into RBCs.** Potential binding partners based on database search and current knowledge in other *Plasmodium* spp. The enzymatic phenotypic profile of each receptor is indicated (R: resistant to pre-treatment; S: sensitive to RBC pre-treatment;? : unknown). The potential effect of blocking agents (i.e. heparin, antibodies) is also displayed. Proteins for which specific antibodies has been tested in this study are highlighted in a box. Other proteins that might have been affected by enzymatic treatment, but for which no antibody was tested, and that might play a role in *P. malariae*-RBC interactions are shown unboxed.

whether *P. malariae* exhibited a clear preference for either CD71+ or CD71- cells. This approach has allowed us to address possible limitations associated with the use of the MT and SG stains, as both methods could potentially identify false positive RBCs. This could occur due to the presence of active mitochondria or nucleic acids in non-parasitized immature reticulocytes that have recently exited the bone marrow. Our findings have pointed, for the first time to our knowledge, that *P. malariae* is capable of invading reticulocytes, although without strict preference for these younger RBCs as observed in *P. vivax* and other non-human *Plasmodium* species. Given that the preference of *Plasmodium* parasites for specific types of RBCs could guide us toward essential receptors required for invasion, it is important to consider the significant differences in the proteomes of reticulocyte and normocyte membranes. For example, certain proteins like CD49d, CD44, and CD98, which has been shown to be used by *P. vivax* [34], are enriched in very immature reticulocytes [72] and these might provide a niche that enhances parasite survival. Capturing these events is difficult due to a combination of factors, such as low parasitemias with reduced number of reticulocytes in the samples and a narrow detection window. These challenges has limited our ability to quantify precisely the frequency of this phenomena in *P. malariae* wild parasites and encourage further studies in this direction to confirm our findings.

Finally, this study has several limitations that should be acknowledged. First, the limited amount of parasite material available hindered, for instance, the use of combinations of different enzymes. However, we mitigated the challenge of a low percentage of new invasions by analysing a large number of events ($10^6$) using the flow cytometer. Another limitation lies in our lack of knowledge of the clonality of the parasite populations. Consequently, the presence of clones with multiple invasion pathways, potentially masking subtle phenotypic differences, cannot be ruled out. Furthermore, the study was limited to a small number of samples, which necessitates additional studies in other geographies with different transmission patterns, to explore deeper into how *P. malariae* penetrates RBCs and whether it affects the maturation of reticulocytes. Such investigations can provide crucial insights for the development of novel antimalarial drugs and vaccines. Additionally, it would be intriguing to explore the potential of the host-cell intracellular environment in triggering sexual commitment in *P. malariae*, as has been demonstrated for *P. vivax* in reticulocytes [73]. Although *P. malariae* is currently not considered a significant threat to malaria elimination, it is imperative to promote research on this parasite to bridge gaps in our understanding of its biological dynamics in circulation. This research can contribute to a more comprehensive knowledge base and inform future strategies for malaria control and eradication.

## Supporting information

**S1 File. Deidentified dataset that was utilized in the analysis.**
(XLSX)

**S1 Fig. A) Representative pictures of intraerythrocytic parasite stages in circulation; B) Transmission electron microscopy image of *P. malariae* late stages; C) Parasitemia at time of collection.**
(TIF)

## Acknowledgments

We acknowledge the imaging facility MRI, and specially Stéphanie Boireau Viala and Myriam Boyer-Clavel, member of the France-BioImaging national infrastructure supported by the French National Research Agency (ANR-10-INBS-04, 'Investments for the future'). We thank the University Clinical Research Center (UCRC) for facilitating access to flow cytometry

instrument and associated analytical software in Mali. We would like to thank all volunteers and families to support our work.

## Author contributions

**Conceptualization:** Alejandro Marin-Menendez, Laurent Dembélé.

**Formal analysis:** Francois Dao, Alejandro Marin-Menendez, Laurent Dembélé.

**Funding acquisition:** Arthur M. Talman, Abdoulaye A. Djimde, Alejandro Marin-Menendez, Laurent Dembélé.

**Investigation:** Francois Dao, Amadou Niangaly, Fanta Sogore, Mamadou Wague, Djeneba Dabitao, Siaka Goita.

**Methodology:** Francois Dao, Amadou Niangaly, Fanta Sogore, Mamadou Wague, Djeneba Dabitao, Siaka Goita, Aboubacrin S. Hadara, Ousmaila Diakite, Mohamed Maiga, Fatoumata O. Maiga, Chantal Cazevieille, Cecile Cassan.

**Project administration:** Cecile Cassan, Alejandro Marin-Menendez, Laurent Dembélé.

**Resources:** Djeneba Dabitao, Arthur M. Talman, Abdoulaye A. Djimde, Laurent Dembélé.

**Supervision:** Arthur M. Talman, Abdoulaye A. Djimde, Alejandro Marin-Menendez, Laurent Dembélé.

**Writing – original draft:** Francois Dao, Alejandro Marin-Menendez, Laurent Dembélé.

**Writing – review & editing:** Francois Dao, Djeneba Dabitao, Arthur M. Talman, Abdoulaye A. Djimde, Alejandro Marin-Menendez, Laurent Dembélé.

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
