## [Decision Letter · Decision Letter 0]

26 Mar 2024

Dear Dr Marin-Menendez,

Thank you very much for submitting your manuscript "Toward elucidating Plasmodium malariae intra-erythrocytic development and invasion using field-isolated parasites in Mali" for consideration at PLOS Neglected Tropical Diseases. As with all papers reviewed by the journal, your manuscript was reviewed by members of the editorial board and by several independent reviewers. In light of the reviews (below this email), we would like to invite the resubmission of a significantly-revised version that takes into account the reviewers' comments. 

We cannot make any decision about publication until we have seen the revised manuscript and your response to the reviewers' comments. Your revised manuscript is also likely to be sent to reviewers for further evaluation.

Sincerely,

Rhoel Ramos Dinglasan

Academic Editor

Susan Madison-Antenucci

Section Editor

Reviewer's Responses to Questions

**Key Review Criteria Required for Acceptance?**

**Methods**

-Are the objectives of the study clearly articulated with a clear testable hypothesis stated?

-Is the study design appropriate to address the stated objectives?

-Is the population clearly described and appropriate for the hypothesis being tested?

-Is the sample size sufficient to ensure adequate power to address the hypothesis being tested?

-Were correct statistical analysis used to support conclusions?

-Are there concerns about ethical or regulatory requirements being met?

Reviewer #1: Objectives are clear. The study is presented as an open-ended characterization with no clear testable hypothesis. To improve this, the authors could add their hypotheses around what they expected to be similar vs. different from P. falciparum/vivax.

Sample sizes are small, but seem reasonable given the challenges of acquiring P. malariae isolates in the field. 

Statistical analyses seem appropriate.

No concerns.

Reviewer #2: In this work, clinical samples were obtained from de-identified donors who had tested positive for P. malariae in a screening study. PCR was used to identify samples with mono infections with P. malariae, which were then investigated for clinical characteristics, parasite phenotypes, and assessment of potential invasion assays. The invasion assays required adoption of a flow cytometry-based method to measure low parasitemia, which has been used in falciparum and vivax studies. The sample sizes were sufficient overall, but in some figures the exact n that was used was not clear, and did not seem to match the number of data points visualized in the figures. Additionally, in several bar graphs it is not clear what is being plotted.

Reviewer #3: The methods are appropriate for this paper and the authors address the limitations in the Discussion due to the use of patient samples.

**Results**

-Does the analysis presented match the analysis plan?

-Are the results clearly and completely presented?

-Are the figures (Tables, Images) of sufficient quality for clarity?

Reviewer #1: Results are clearly presented. I would suggest making figure titles more specific, indicating a clear conclusion rather than general characterization. Some specific suggestions for figures, particularly Figure 4, are below. I would suggest including exact p-values, not just asterisks. 

I would suggest substantial revisions to Figure 6. The current figure seems more appropriate to a review article. Rather than summarizing which ligands are susceptible to the different enzymes according to the literature, I think it would be more useful to make a model with proposed invasion pathways for P malariae based on the results in this paper.

Reviewer #2: (No Response)

Reviewer #3: Yes the results are presented clearly and the figures are of sufficient quality for clarity.

**Conclusions**

-Are the conclusions supported by the data presented?

-Are the limitations of analysis clearly described?

-Do the authors discuss how these data can be helpful to advance our understanding of the topic under study?

-Is public health relevance addressed?

Reviewer #1: Conclusions are more negative results than positive, but are important contributions to the field given the large gap in knowledge in this area. 

The discussion seems to imply a bigger importance for reticulocytes than the results suggest- I would de-emphasize this a bit and discuss more the implications/hypotheses underlying why reticulocyte invasion is observed in a small number of individuals but not majority. 

Why is there no impact of CD35, CD147, CD235a antibodies if these are sensitive to 1 or more of the enzymes and there were significant differences with the enzymes? This point, and more generally the integration of the antibody and enzyme data, should be discussed in Discussion. 

A major limitation, which the authors address, is the low percentage of new invasions and therefore difficulty in assessing significant changes. The decision to count a higher number of events as a result seems reasonable. 

The public health relevance could be highlighted more. For example, the impact of anemia on morbidity could be mentioned, and the incidence of P malariae in Mali could be discussed.

Reviewer #2: (No Response)

Reviewer #3: Yes the conclusions are supported by the data presented. There are some caveats to using monoclonal antibodies against red blood receptors to examine P. malariae invasion and I have highlighted this to be addressed in the discussion.

**Editorial and Data Presentation Modifications?**

Reviewer #1: The title seems a little awkward and not as compelling as it could be. I would suggest something like “Malian field isolates provide insight into Plasmodium malariae intra-erythrocytic development and invasion”

Line 61 – I would replace “although the underlying mechanism are not yet known” with “leading to substantial impacts of morbidity” or something that highlights the clinical impact of glomerulonephritis/anemia 

Clarify in methods that 45 individuals recruited were positive for P. malariae (not just Plasmodium-infected as it says in line 138)

Clarify inclusion criteria – is diagnosis of P. malariae by blood smear? Or qPCR?

Clarify complete media components. Are culture conditions similar to Pf? Are these conditions well-established for P. malariae? 

Add antibody dilutions used

Line 286 – what does the existence of knobs suggest about possible surface ligands? 

Figure titles – change to make more specific and state the conclusion of figure, e.g. for figure 2, “P. malariae replicates and matures in vitro” instead of “Characteristics of P. malariae in vitro”

Line 319 – enter not entry

Line 327 – latter not later 

Line 330- I would use phrase other than a “battery”

Figure 2 - does n=178 refer to individual cells? how many patients is this from?

Figure 3 - are cells gated at all prior to SG/CTFR gate shown? It would be worth having a singlet gate to confirm you are not just measuring autofluorescence. 

Figure 3C- include N. 

Figure 4 – It doesn’t seem necessary to me to include individual graphs and to show the Control at 100% so many times. I would recommend including all antibodies on the same graph, removing the control dots, and indicating on the axis that the values are normalized. 

Figure 4C- what are the gray dots in the background? can these be removed?

Figure 5B – why is it needed for quadrant gates to not be straight? Is the compensation definitely correct? 

Figure 5C – the green dot doesn’t look that different from 0 – is this confirmed reinvasion? It looks to me like only the red dot is clearly invading reticulocytes. 

Discussion – wording and sentence structure could be improved generally. 

Figure 6 – cite at end of Results section. 

Line 402 – expand more on fam-l and fam-m genes. Also clarify why you expect a homolog of Rh5 to be more likely to be important rather than a new unidentified receptor. Including experiments that target these proteins would definitely make the paper significantly stronger, but I realize that creating new antibodies may be prohibitive.

Line 433 – Rephrase – for example “furthermore, the study was limited to a small number of samples, which necessitates additional studies…”

Line 436 – “antimalarial drugs” instead of “antimalarials”

Reviewer #2: 1. The information on the parasite morphologies and life cycle stages identified in the samples adds to our knowledge. However it would also be helpful to have some information about the parasitemia in these samples. This should ideally be added as a panel to Figure 1.

2. Given the low parasitemias observed in Figure 3C, for Figure 4 the raw parasitemia data should be made available in a supplemental file to assist with understanding the biological significance. 

3. For the invasion assays, what was the initial starting parasitemia that was added?

Reviewer #3: Required changes:

1. In Table 2 and Figure 6, it is stated that the enzyme treatment requirement for PvRBP2b is not known. This is incorrect as in Fig. S1 of Gruszczyk et al Science 2018, it was shown that PvRBP2b binding to reticulocytes was sensitive to trypsin and chymotrypsin but resistant to neuraminidase. This information should be updated in Figure 6, Table 2 and all the corresponding text within the manuscript. 

2. In Table 2, CD71 is referred to as Transferrin. This is incorrect and CD71 should be referred to as Transferrin Receptor 1. 

3. In Table 2, for PfRh4, please also add reference Tham et al, Infection and Immunity, 2009 PMID: 19307208 as this was the first paper to describe the enzyme profile for the binding. 

4. Line 95 to 96 in Introduction states “Secondly, the P. vivax RBP2b that interacts with one of the key reticulocyte markers,the transferrin receptor (CD71)(32)(33)(34)(35).” Most of the references for this statement are incorrect and should be changed to (Gruszczyk et al Science 2018 PMID: 29302006, Gruszczyk and Huang et al Nature 2018 PMID: 29950717, Chan et al, Nature Communications 2021PMID: 33750786, Chan et al, Cell Microbiology 2020, PMID: 31469946)

5. In Introduction, line 103 to 105, “One of those, the RH5 kite-shaped fold, is known to be also present in RBP2a in P. vivax and may be a key conserved structure for the binding capabilities of all Rh and RBP genes (41). Please also add Gruszczyk et al Science 2018 and Gruzczyk and Huang et al, Nature 2018 as references as PvRBP2b also shows the same structural fold as PfRh5. 

Minor consideration:

1. Under Molecular Players in P. malariae RBC invasion: The authors provide a summary of potential orthologues based on existing genome annotation of Plasmodium genomes. It would be interesting to run an Alphafold/Foldseek comparisons of the standard structural scaffolds of PvRBP/PfRh family and PfEBA family of proteins to observe what is present structurally in the P. malariae genome. This is only a suggestion and if not done, it does not impact the novelty of this paper. 

2. More recently, CD98 was observed to be a receptor used by P. vivax potentially through binding of PvRBP2a. Can this be updated in the relevant tables and figures? No required additional experiments. 

3. Add an additional reference to “Second, the reticulocyte binding protein (RBP) family that includes the P. yoelii 235-kDa rhoptry proteins, the P. vivax reticulocyte binding proteins (PvRBP-1 and -2), and the P. falciparum reticulocyte homology proteins (PfRH1, PfRH2a, PfRH2b, PfRH4, and PfRH5) (25)(24)”. Please add Chan et al, 2020 Cell Microbiology PMID: 31469946 as it also provides a more up to date review of the PvRBP family of proteins. 

4. For Figure 2, if already available, would it be possible to also have figure panels showing the presence of rings at 96 hr to show that there has been a reinvasion process to go together with the flow cytometry plots?

To be addressed in Discussion:

The authors use a panel of antibodies against known receptors and show that the addition of those antibodies does not block P. malariae invasion. From the panel chosen, many of the antibodies are monoclonal antibodies that have not been shown to disrupt the known invasion ligand-receptor interaction or parasite invasion. For example, anti-CD35 (BioLegend, 332402) and anti-CD71 (Miltenyi Biotec, 130-124-327 (Clone REA902) have never been shown to block P. falciparum or P. vivax invasion. In particular, there is no anti-CR1 mAb known to block invasion and also only 5 out of 9 anti-CD71 mAbs tested in Gruszczyk et al Science 2018 were shown to block reticulocyte binding. Therefore, it would good to address these caveats in the Discussion and propose some ways forward for future studies that do not include untested monoclonal antibodies:

a. Most monoclonal antibodies chosen in this paper have not been shown to block P. falciparum and P. vivax invasion.

b. May be important to initially use polyclonal antibody against the receptors if the binding sites for P. malariae is not known.

c. If available, the use of recombinant soluble receptors such as soluble CR1 (Tham et al, PNAS 2010) to understand the effect of inhibitor.

**Summary and General Comments**

Reviewer #1: This study uses Plasmodium malariae field isolates to characterize the invasion and development of this parasite. The authors effectively put their work into the context of what is known about intraerythrocytic development of P falciparum and P vivax. They develop an assay to assess in vitro invasion and development by the parasite, and to probe the role of different receptors of interest. Overall, given that P malariae is an understudied parasite with very little known about its biology, the contributions by Dao et al. are novel and valuable to the field. The manuscript could be strengthened with some edits to Figures 4 and 6 and editing the language to be clearer.

Reviewer #2: In their study, Dao et al investigate clinical samples of Plasmodium malariae, an under-studied species of Plasmodium parasites that can cause malaria in humans. The lack of an in vitro culture system for P. malariae and its tendency to occur as a co-infection have limited its use in experimental studies. The demonstration of an assay to study invasion of P. malariae from clinical isolates in vitro is an important advance for our understanding of this organism. 

1. A primary finding from the analysis of the clinical characteristics of the samples was that P. malariae is associated with anemia. However the data in Table 1 do not provide any indication of the level of anemia in the general population (not infected with P. malariae). Given that anemia is often multi-factorial, the prevalence of anemia in the villages/age groups from which the samples were drawn should be reported. 

2. For several figures, there is not a clear indication of the sample number used and how it relates to the presented data. For Figure 1A, what is the sample number and what do the error bars represent? 

For Figure 2A, what is meant by n=178? The study reportedly only used 8 samples for in vitro culture – please explain n=178. Similarly, why is n=65 for Figure 2C? Is this data from 65 schizonts from a single donor? Please clarify.

For the invasion assays presented in Figure 3C, please include the sample number (n). The text implies that parasitemias were quantified in duplicate, but what is plotted on the graph in 3C? 

In 4A, legend indicates n=3, but what is this number referring to? Most of the graphs seem to show 4-5 data points. 

In Line 339, percent reduction in invasion is listed (referring to Figure 4B). Please clarify what these numbers refer to: is it the mean reduction for the 6 samples?

3. For figure 1A, what is the explanation for the low percentage of rings reported for the samples, when Figure 2A demonstrates a high percentage of rings at 0 hours? 

4. In Figure 4, 8 different antibodies are tested for their ability to block P. malariae invasion. Are these the same clones/concentrations that have been shown to impact Pf/Pv invasion in the literature? This would be important information to support the conclusion that P malariae entry may not rely on any of these receptors. 

5. The conclusion that P. malariae is capable of invading reticulocytes would be important, but the evidence supporting this conclusion is not sufficient because Figure 5 is difficult to interpret. At a minimum, the flow cytometry plots from the two samples reported to demonstrate invasion in reticulocytes should be shown. However the conclusion would be better supported by using FACS to sort SG+/CD71+ cells to show that these are indeed infected by P. malariae.

Reviewer #3: The manuscript by Dao et al describes characteristics of P. malariae infection/invasion from patient samples and a one cycle culture assay which they have established. P. malariae is one of the six malaria parasites that infect humans and is very understudied. As global efforts move towards malaria elimination, it is paramount that the basic biology of how P. malariae infects human red blood cells is understood. As there is no existing culture system for P. malariae, the authors are using patient samples which provide an important snapshot of what is occurring in human infections. In addition, to the best of my knowledge, the authors also provide evidence for an in vitro short-term culture that shows at least one cycle of reinvasion which is an important advance in the field. They proceed to determine if the receptors, surface expression properties (via enzyme treatment of RBCs) and types of RBCs are similar between what is known for P. vivax and P. falciparum. This is an impressive line of questioning in light that it has been done with patient samples and therefore this paper provides some important insights into P. malariae invasion.

PLOS authors have the option to publish the peer review history of their article (what does this mean? ). If published, this will include your full peer review and any attached files.

**Do you want your identity to be public for this peer review?** For information about this choice, including consent withdrawal, please see our Privacy Policy .

Reviewer #1: Yes: Kathleen Dantzler Press

Reviewer #2: No

Reviewer #3: Yes: Wai-Hong Tham
---

## [Decision Letter · Decision Letter 1]

25 Jul 2024

Dear Dr Marin-Menendez,

Thank you very much for submitting your manuscript "Malian field isolates provide insight into *Plasmodium malariae*  intra-erythrocytic invasion and development" for consideration at PLOS Neglected Tropical Diseases. As with all papers reviewed by the journal, your manuscript was reviewed by members of the editorial board and by several independent reviewers. In light of the reviews (below this email), we would like to invite the resubmission of a significantly-revised version that takes into account the reviewers' comments. 

We cannot make any decision about publication until we have seen the revised manuscript and your response to the reviewers' comments. Your revised manuscript is also likely to be sent to reviewers for further evaluation. We urge the authors to meticulously re-examine all of the numbers in the text and figures and consider all of the points raised by reviewers. A colleague who can take a fresh look at the manuscript may be helpful in this regard.

Sincerely,

Rhoel Ramos Dinglasan

Academic Editor

Susan Madison-Antenucci, PhD

Section Editor

Reviewer's Responses to Questions

**Key Review Criteria Required for Acceptance?**

**Methods**

-Are the objectives of the study clearly articulated with a clear testable hypothesis stated?

-Is the study design appropriate to address the stated objectives?

-Is the population clearly described and appropriate for the hypothesis being tested?

-Is the sample size sufficient to ensure adequate power to address the hypothesis being tested?

-Were correct statistical analysis used to support conclusions?

-Are there concerns about ethical or regulatory requirements being met?

Reviewer #1: No major comments. I did wonder about the single concentration chosen for the antibodies being tested. Why was this concentration chosen and is it possible you would have seen different results if a different concentration was used? If this concentration has been used by other studies, those should be cited in the methods.

Reviewer #2: The objectives of the study are clearly articulated, and the study design using patient samples is appropriate to investigate the biology of P. malariae since this organism cannot be readily cultured. Sample sizes seem appropriate, though there are inconsistencies in the sample size reporting, which should be corrected.

**Results**

-Does the analysis presented match the analysis plan?

-Are the results clearly and completely presented?

-Are the figures (Tables, Images) of sufficient quality for clarity?

Reviewer #1: The figures and discussion of results are much improved. The figure/section headings and inclusion of p-values and sample sizes are also improvements. My one remaining concern is Figure 6. The accompanying paragraph describing possible hypotheses for invasion is an interesting discussion of the implications and knowledge gaps associated with this work. However, the figure does still imply that a lot more is known or backed up with evidence than the results from this paper indicate. The explanation of possible redundancy across pathways makes sense but that still doesn’t provide any clear support for those proteins being definitively involved. Could you adjust the figure so that it 1) does not include proteins where blocking did not impact invasion in this paper (unless you have strong reason to believe this was because of redundant pathways) and 2) highlights proteins that have known sensitivity to the enzymes but were not tested in this paper with question marks ? Alternatively, since this is more speculative than backed up with data, it might also be worth considering removing the figure and moving the hypothesis paragraph to the discussion.

Reviewer #2: The figures and analysis presented match the analysis plan of the study. However, there are several aspects of figures that are unclear, and sometimes there is inconsistency between the text, legends, and/or methods.

1. For Figure 1B, the “n” should be 23, not 11, because data from all 23 samples are plotted. 

2. In Figure 2, the number of samples used is still not clear. The results section indicates that 8 patient samples were used, but the legend says n=23.

3. In Figure 2A, the authors show the changes in parasite stages over time in vitro culture, but there is no information about whether the percentages presented are additive for all samples, average, or something else. This does not allow for any information about the variability between samples. A more informative visualization of these data would be to plot the data for each sample, with points for each stage (represented by different shapes) and each sample. Thus, at each time point a total of 23 x 3= 69 points should be plotted (or 8x3= 24 points if indeed the sample number was 8 and not 23). Alternatively, the data could be plotted as a bar graph as in current version, but the data included data should be from each sample or average for all samples (with standard number of cells counted per sample indicated), and error bars/ standard deviation should be provided.

4. In response to a concern about the discrepancy between the ring stage parasitemias shown in Figure 1A and Figure 2A, the authors suggest in their response that the samples in Figure 2A were magnet purified, which might enrich for rings. However, there is no indication of magnet purification having been performed for the data presented in Figure 2 in the methods, legend, or results sections. Indeed, the experimental method implies that samples went straight from isolation to culture without any manipulation beyond washing. This needs to be reviewed and clarified. 

5. In Figure 2B, the legend indicates NV, R, T, and S, but these notations are not found in the revised figure.

6. In Figure 3C, the legend indicates n=18, but the text in the results section indicates that the sample number was 9 and that each sample was quantified in duplicate with the average parasitemia plotted. The graph itself in figure 3C appears to show somewhere between 10-18 data points (~16 lines connecting the two timepoints are seen). Please correct these inconsistencies in the data, text, and legend.

Reviewer #3

Figure 1 B) the original figure listed n=28. The figure legend now states n=11 but there are clearly more data points in figure B. It appears that the 12 specimens had no shizonts. To be consistent the n should be listed as 23 to include the samples that had no schizonts as is the case in panel C where the samples with no gametocytes are included in the “n” number.

Figure 2A) the figure lists 23 samples and n=121 for smears. If a smear was done for each sample at all 7 time points the number of smears should be 161. Further confusion is caused by the text which states that “To investigate the dynamics of parasite development, we used eight patient samples, incubated them under standard culture conditions and monitored them every 12h using light microscopy. This must be clarified.

In Figure 2B, the legend indicates NV, R, T, and S, but these notations are not found in the revised figure.

In Figure 3C, the legend indicates n=18, but the text in the results section indicates that the sample number was 9 and that each sample was quantified in duplicate. Does the plot show the each of the duplicates. Please clarify.

Please note that line numbers given in the response to reviewers do not match the text of the revised manuscript which complicated the review. 

**Conclusions**

-Are the conclusions supported by the data presented?

-Are the limitations of analysis clearly described?

-Do the authors discuss how these data can be helpful to advance our understanding of the topic under study?

-Is public health relevance addressed?

Reviewer #1: The discussion (including description of limitations, relationships between Pm and other Plasmodium species and broader significance) is significantly improved from the previous version.

Reviewer #2: The conclusions are supported by the data presented. The discussion section nicely laid out the strengths and limitations of the study, as supported by the data and the current state of the literature.

**Editorial and Data Presentation Modifications?**

Reviewer #1: The language clarity has been significantly approved. Some additional minor comments:

Introduction – the order of sentences feels a little confusing especially given that the manuscript is focused on Pm– I would suggest the first sentence identifying Pm as important, then introduce all 5 Plasmodium species, then highlight Pf and Pv as responsible for most cases/deaths, then get to importance of Pm

Line 57 – sentence starting with “secondly” is not a full sentence. Suggest “Secondly, P. falciparum parasites invade and grow in RBCs of all maturities and their development ends with bursting the RBCs which can lead to severe anaemia.”

Line 80 – similar issue to line 57

Line 117 – “quantitative” instead of “powerful”

Line 120 – add “of” after “absence”

Line 285 – is this average number merozoites? Median? 

Figure 1B and 2C– axes are confusing. I would abbreviate “number” rather than “merozoite” (e.g. “# merozoites per schizont”). 

Line 301 – change “that informed of the viability of the parasites” to “to measure parasite viability”

Line 312 – change “oppositely” to “in contrast” 

Discussion – There seems to be a lot of variability between sample response to antibodies – is it possible that some samples are responding to the antibody?

Line 369 – remove “of” after “despite”

Line 375 – 377 – this sentence is confusing and doesn’t seem necessary

Line 382-383 change to “No var genes orthologues have been identified, unlike in P. falciparum where PfEMP1s in the RBC membrane are encoded by var genes”(59)(60).

Line 418 - remove “of” after “despite”; change the “of” after “role” to “for”

Line 421 – change “then “to “thus”

Line 449 – change “has rendered” to “is”

Line 460-461 – also mention the need to investigate samples in other geographies with different transmission patterns

Reviewer #2: Please see above.

**Summary and General Comments**

Reviewer #1: Overall, this manuscript is significantly improved. I would recommend acceptance following changes to Figure 6 (either modification or removal of figure and move of accompanying text to discussion) and have suggested some additional minor improvements. The manuscript will provide useful information to the field about the biology of Plasmodium malariae, an under-characterized yet important malaria parasite.

Reviewer #2: (No Response)

PLOS authors have the option to publish the peer review history of their article (what does this mean? ). If published, this will include your full peer review and any attached files.

**Do you want your identity to be public for this peer review?** For information about this choice, including consent withdrawal, please see our Privacy Policy .

Reviewer #1: Yes: Kathleen Dantzler Press

Reviewer #2: No
---

## [Decision Letter · Decision Letter 2]

17 Oct 2024

Dear Dr Marin-Menendez,

Thank you very much for submitting your manuscript "Malian field isolates provide insight into *Plasmodium malariae*  intra-erythrocytic invasion and development" for consideration at PLOS Neglected Tropical Diseases. As with all papers reviewed by the journal, your manuscript was reviewed by members of the editorial board and by several independent reviewers. In light of the reviews (below this email), we would like to invite once again the resubmission of a significantly-revised version that takes into account the reviewers' comments, especially with respect to the claims. 

We cannot make any decision about publication until we have seen the revised manuscript and your response to the reviewers' comments. Your revised manuscript is also likely to be sent to reviewers for further evaluation.

Sincerely,

Rhoel Ramos Dinglasan

Academic Editor

Susan Madison-Antenucci

Section Editor

Reviewer's Responses to Questions

**Key Review Criteria Required for Acceptance?**

**Methods**

-Are the objectives of the study clearly articulated with a clear testable hypothesis stated?

-Is the study design appropriate to address the stated objectives?

-Is the population clearly described and appropriate for the hypothesis being tested?

-Is the sample size sufficient to ensure adequate power to address the hypothesis being tested?

-Were correct statistical analysis used to support conclusions?

-Are there concerns about ethical or regulatory requirements being met?

Reviewer #1: no additional comments

Reviewer #4: The objectives of the study are clearly outlined and the methodology employed adequate, as are the materials used. The sample sizes are adequate to address some but not all of the hypothesis tested. There are no specific concerns for the statistical analyses and none for the ethical or regulatory requirements.

Overall, some crucial aspects of the methodology are poorly described or not provided. These must be included in any revised version (see detailed comments below)

Introduction line 82: P. vivax encodes 11 members of the RBP gene family not only 2.

Line 97: Reference for CD98 as P. vivax reticulocyte receptor is missing (Malleret B, et al.. Plasmodium vivax binds host CD98hc (SLC3A2) to enter immature red blood cells. Nat Microbiol. 2021 Aug;6(8):991-999.)

Line 112: the review (reference #44) does not mentioned that “P. berghei invade reticulocytes 150 times more than normocytes”. The following reference mentioned that PbA was 5.2 times more likely to invade a reticulocyte than a normocyte (Leong YW, et al.. Rodent Malaria Erythrocyte Preference Assessment by an Ex Vivo Tropism Assay. Front Cell Infect Microbiol. 2021 Jul 12;11:680136.)

Line 119: Add references for Flow cytometry and parasitemia/cell tropism monitoring.

It will be useful to describe briefly the culture medium used for P. malariae in vitro culture, including the type of serum, antibiotics…..

Line 191: What is the average yield of P. malariae after magnetic sorting.

Line 215: Effect of antibodies in invasion. The authors must use Fab’ fragment antibody to avoid steric hindrance of the antibodies during the P. malariae invasion inhibition assay. The antibodies must also be azide-free.

Line 242: What is the source of immature reticulocyte for the reticulocyte invasion assay? Do the authors enriched the reticulocytes from cord or peripheral blood? If the authors did not add any reticulocytes, it is not an reticulocyte invasion assay but simply a flow cytometry phenotyping to identify CD71 positive Pm-infected red blood cells.

**Results**

-Does the analysis presented match the analysis plan?

-Are the results clearly and completely presented?

-Are the figures (Tables, Images) of sufficient quality for clarity?

Reviewer #1: Figure 4B: I would suggest showing the control rather than normalizing to it, so that we can visualize how much the enzymes decreased invasion. 

The edits to Figure 6 and its placement within the discussion are much improved and feel more consistent with the findings of this study.

Reviewer #4: There are some concerns with respect to the quality of some of the data/images, as noted below

Lines 281-290: It is not clear from the rather image provided that the authors have observed merozoites. From the Giemsa stained picture, as well as from the EM picture, I can only see nuclear masses, NOT fully formed merozoites. The typical mature schizont is characteristically in the form of a central large pigment with the merozoites arranged radially around it. Have the authors seen any of these forms? Whereas the authors have observed some protrusions on the red cell surface, they are suggestive of some membrane “decoration” but not of knobs (as is understood for P. falciparum). Thus in here and in the Discussion, the word “putative” should be added before the word “knob”

Figure 1A: The authors need to provide the parasitemia at time 0 for each sample before showing the parasite staging. A supplementary Figure should show the Giemsa staining of the different samples. 

Figure 1D: The TEM images are too small to be informative. The authors must show rings and trophozoites to validate the presence of membrane protrusions at the surface of schizonts. It is premature to call them knobs for P. malariae.

Lines 292-301: The interpretation of the evolution of the parasites in the cultures is not necessarily consistent with the observations/data presented. It is highly likely that the authors observed a single invasion event, albeit for only a minor portion of the parasites, but it is likely that a majority of the parasites were not maturing. Again it would be important to show and count the number of mature schizonts with fully formed merozoites.

Figure 2A. The in vitro culture of the negative fraction of P. malariae magnetic sorting show a range of trophozoite between 62-82% within 96 hours. Surprisingly the highest level of reinvasion was observed at 48 hours. The authors need to show many examples of the Giemsa-stained parasites at the different time points of the in vitro culture to show the morphology and viability of the parasites.

Figure 2B. The percentages for the events in each gate are missing. The author must show the staging follow-up by flow cytometry with a histogram as presented in Figure 2A for microscopy. Between time 0 and 12 hours the main population of trophozoites disappears meaning that the parasites died because the schizont gate is totally empty.

Page 328-345: The very low starting parasitaemias close to the limit of detection by cytometry makes it difficult to interpret the comparative invasion/inhibition assays (Fig3 and Fig4).

Figure 3A. What is the source of uninfected red blood cells added for the invasion assay? If they are healthy donors, are they part of the IRB Nº2020-2023/168/CE/FMPOS/FAPH?

Figure 3B. The percentages in the flow cytometry gates must be added.

Figure 3C. What is the limit of detection for infected red blood cells with the flow cytometry assay when the author used uninfected red blood cells as negative control? The parasitemia at 12h are around 0.01% and it is very close to the limit of detection for flow cytometry due to artefact signal associated with reticulocytes, leukocytes or dead parasites with Sybr Green (DNA staining) and a Blue laser for excitation.

Line 347-354: This is the weakest part of the experimental data. The authors cannot conclude with this set of experiments that P. malariae can infect reticulocytes, especially as this was only found for a single sample (out of 8). Thus, in the Abstract line 31 in the abstract should be removed, and words “of different maturities” in line 33, and “and that it can invade red blood cells of different all maturities” in the Author Summary.

Lines 433-452: Given the above comments and those below the claim that this is the first demonstration of reticulocyte invasion by P. malariae should be removed, as it is very poorly supported by data that at best hints at this possibility. Chwatt (Ref 48 of the manuscript) had described this as an “exceptional occurrence”. The data presented to support the claim might well result from a an artefact of the technique.

Figure 5. To claim any reticulocyte invasion the authors need to add reticulocytes and monitor the expression of CD71 at the surface of the pre-stained reticulocytes before and after invasion. For the only P. malariae sample displaying CD71+ Pm-infected RBC, the signal detected with Sybr Green and CD71 PE in the CD71 high population is due to highest autofluorescence of the immature reticulocytes. It is the reason why this percentage is not found with the CD71 APC. 

Figure 5C. The author must explain the calculation done to obtain the percentage of infected reticulocytes.

**Conclusions**

-Are the conclusions supported by the data presented?

-Are the limitations of analysis clearly described?

-Do the authors discuss how these data can be helpful to advance our understanding of the topic under study?

-Is public health relevance addressed?

Reviewer #1: no additional comments

Reviewer #4: The authors provide an interesting set of observations aimed at illuminating some aspects of the biology of the erythrocytic stages of P. malariae. This parasite species is little studied and thus, the date is of interest to the community.

My main, and major concern, with the twice-revised manuscript are the rather emphatic claims that the authors make (optimized culture conditions, significance of invasion assays, and most glaringly, definitive demonstration of reticulocyte invasion) that are supported by observations on a few samples, furthermore using conditions of parasite development that are far from ideal, and on techniques that are close to their limits of detection, principally because of the low parasitaemias that are inherent to P. malariae clinical samples. Moreover, important technical data (such as schemes for calculations) are missing. This is tacitly implied in the text and some of the limitations are explicitly stated in the Discussion.

**Editorial and Data Presentation Modifications?**

Reviewer #1: Lines 79, 81 and 96:

Remove “that” to make it complete sentence 

Line 164: “every” not “each”

Line 326: “Evaluate”, not “evaluating”

Line 341: remove “allowed us to quantify all three enzymes”

Line 353: replace “was confirmed with the two approaches used” with something more specific, like “showed invasion of reticulocytes” 

Line 395-396

This sentence feels very long. I would edit to “…it is likely to employ alternative mechanisms to gain entry into RBCs, as well. A model showing potential invasion pathways is shown in Figure 6.”

Line 396

I would modify this sentence to “In alignment with this hypothesis, a report from the last decade described an unconventional invasion pathway utilized by Colombian and Peruvian field isolates that was resistant to simultaneous pre-treatment of RBCs with these enzymes.”

Line 403: change “presence” to “identity”

Line 424-425: remove “displaying also the effects of potential blocking agents”

Line 441: remove “Contrary to existing knowledge,”

Line 449: “parasite” not “parasite’s”

Line 449: remove “Despite our findings” and break up sentence into 2: “Capturing these events is difficult due to a combination of factors, such as low parasitemias with reduced number of reticulocytes in the samples and a narrow detection window. These challenges have limited our ability to quantify precisely the frequency of this phenomena in P. malariae wild parasites.”

Reviewer #4: (No Response)

**Summary and General Comments**

Reviewer #1: Manuscript is significantly improved from the previous versions!

Reviewer #4: The data presented in this manuscript are of interest, but these are insufficient to support some of the conclusions and claims presented (see above).

PLOS authors have the option to publish the peer review history of their article (what does this mean? ). If published, this will include your full peer review and any attached files.

**Do you want your identity to be public for this peer review?** For information about this choice, including consent withdrawal, please see our Privacy Policy .

Reviewer #1: Yes: Kathleen Dantzler Press

Reviewer #4: No
---

## [Decision Letter · Decision Letter 3]

16 Dec 2024

Dear Dr Marin-Menendez,

We are pleased to inform you that your manuscript 'Malian field isolates provide insight into *Plasmodium malariae*  intra-erythrocytic invasion and development' has been provisionally accepted for publication in PLOS Neglected Tropical Diseases.

Best regards,

Rhoel Ramos Dinglasan

Academic Editor

Susan Madison-Antenucci

Section Editor

Shaden Kamhawi

co-Editor-in-Chief

Paul Brindley

co-Editor-in-Chief

Reviewer's Responses to Questions

**Key Review Criteria Required for Acceptance?**

**Methods**

-Are the objectives of the study clearly articulated with a clear testable hypothesis stated?

-Is the study design appropriate to address the stated objectives?

-Is the population clearly described and appropriate for the hypothesis being tested?

-Is the sample size sufficient to ensure adequate power to address the hypothesis being tested?

-Were correct statistical analysis used to support conclusions?

-Are there concerns about ethical or regulatory requirements being met?

Reviewer #4: (No Response)

**Results**

-Does the analysis presented match the analysis plan?

-Are the results clearly and completely presented?

-Are the figures (Tables, Images) of sufficient quality for clarity?

Reviewer #4: (No Response)

**Conclusions**

-Are the conclusions supported by the data presented?

-Are the limitations of analysis clearly described?

-Do the authors discuss how these data can be helpful to advance our understanding of the topic under study?

-Is public health relevance addressed?

Reviewer #4: (No Response)

**Editorial and Data Presentation Modifications?**

Reviewer #4: (No Response)

**Summary and General Comments**

Reviewer #4: The authors have responded adequately to the reviewers' concerns

PLOS authors have the option to publish the peer review history of their article (what does this mean? ). If published, this will include your full peer review and any attached files.

**Do you want your identity to be public for this peer review?** For information about this choice, including consent withdrawal, please see our Privacy Policy .

Reviewer #4: No

---

## [Editor Report · Acceptance letter]

Dear Dr Marin-Menendez,

We are delighted to inform you that your manuscript, "Malian field isolates provide insight into *Plasmodium malariae*  intra-erythrocytic invasion and development," has been formally accepted for publication in PLOS Neglected Tropical Diseases.

Best regards,

Shaden Kamhawi

co-Editor-in-Chief

Paul Brindley

co-Editor-in-Chief
